# Reducing Automotive Cooling System Complexity through an Adaptive Biomimetic Air Control Valve

**DOI:** 10.3390/biomimetics9040207

**Published:** 2024-03-29

**Authors:** Thomas Thuilot, Moses-Gereon Wullweber, Matthias Fischer, Michael Bennemann, Tobias Seidl

**Affiliations:** 1Westfalian Institute for Biomimetics, Westfalian University of Applied Sciences, Münsterstraße 265, 46397 Bocholt, Germany; thomas.thuilot@hs-ruhrwest.de (T.T.); moses-gereon.wullweber@w-hs.de (M.-G.W.); matthias.fischer@w-hs.de (M.F.); webmaster@michael-bennemann.de (M.B.); 2Institute of Measurement Engineering & Sensor Technology, Ruhr-West University of Applied Sciences, Duisburger Straße 100, 45468 Mülheim an der Ruhr, Germany; 3Fachrichtung Physik, Universität des Saarlandes, Campus E2.6, 66123 Saarbrücken, Germany; 4Institute of Informatics, Hochschule Ruhr West, Lützowstrasse 5, 46236 Bottrop, Germany

**Keywords:** pit valve, cooling system, bordered pit, pressure-controlled, compliant

## Abstract

Future automotive mobility is predominantly electric. Compared to existing systems, the requirements of subsystems change. Air flow for cooling components is needed predominantly when the car is in rest (i.e., charging) or at slow speeds. So far, actively driven fans consuming power and generating noise are used in this case. Here we propose a passive adaptive system allowing for convection-driven cooling. The developed system is a highly adaptive flat valve derived from the bordered pit. It was developed through an iterative design process including simulations, both structural and thermodynamic. In hardwoods and conifers, bordered pits enable the challenging transport of vertical fluids by locally limiting damage. Depending on the structure, these can close at sudden pressure changes and take the function of valves. The result of the biomimetic abstraction process is a system-integrative, low-profile valve that is cheap to produce, long-lasting, lightweight, maintenance-free, and noise-free. It allows for the passive switching of air flow generation at the heat exchanger of the cooling between natural convection or an active airstream without the need for complex sensing and control systems. The geometric and material design factors allow for the simple tuning of the valve to the desired switching conditions during the design process.

## 1. Introduction

### 1.1. Cooling in Electric Mobility

The ways we drive are changing: with 31.4% of all newly registered cars in Germany in 2022 being purely electric, automotive mobility is on the rise [1]. Even in terms of fluctuations, individual automotive transport is clearly shifting away from carbon-fuel-powered propulsion towards battery-powered electric propulsion [2,3,4]. With this come changes in packaging but also changing demands on secondary aggregates. Generally, they should be more silent since no combustion engine masks their noise, and should consume little power to extend range [5,6]. Preferably, they should reduce the system complexity and parts required for cost (and weight) optimization. Specifically, they need to fulfill different requirements during operation and hence need to be tuned accordingly [7]. Combined, decades of fine-tuning fuel- and combustion-based automotive design needs to be repeated for battery–electric design to reach a similar high degree of development.

One primary issue is thermal management, predominantly the dissipation of waste heat, i.e., cooling [8,9]. In fuel combustion cars, heat production predominantly correlates with the duration of use and the velocity attained, and hence heat dissipation is mostly achieved by harvesting the air motion resulting from self-motion. This source of cooling is strong enough that current cooling systems are designed to generate a strong back pressure and only let a little air pass through the system while maintaining the best possible aerodynamic properties of said vehicle. Only in rare moments of overheating (i.e., stop-and-go traffic, end of travel) added active ventilation is required to generate additional air flow. Noise emissions and power consumption are accepted as a rare nuisance.

On the other hand, battery–electric vehicles generate the most heat both during charging and at low speeds when generating high torque (e.g., uphill driving) [8,9]. In both cases, “free” wind cannot be harnessed, and additional generated ventilation becomes a recurring, noisy, and power-consuming nuisance.

The next obvious solution—an active air flow management that identifies driving state and cooling demand and actively opens (and closes) airways to allow for energy- and noise-free flowing air and hence heat exchange—would increase system complexity and negatively impact energy consumption [10,11]. In the present account, we report on the development of a novel, bioinspired, cost-effective, parameterizable, and silent valve that allows for adaptive air flow management switching between “classic” air flow for high speeds, thus retaining the important aerodynamic properties of the car and allowing for “free” uninhibited air flow through the cooling system at low driving speeds. The behavior and structure of the bordered pit—commonly found between the lignified cell walls of higher plants—have been used as a functional model.

### 1.2. Biological Model: Bordered Pit

Pits are an essential functional structure for the water and material transport of higher plants. The bordered pit is a flat valve that is found in angiosperms and gymnosperms at the junction between two cells. Depending on the type, they range in size between 4 and 10 μm. They connect cells, unifying their internal medium and allowing for the free exchange of liquid and nutrients. After a sudden loss of pressure in one cell, the bordered pits instantaneously close and prevent neighboring unharmed cells from collapsing [12,13]. In gymnosperms, closing happens mostly in the pressure range from 1 MPa to 12 MPa [14].

In more detail, a pair of pits, which are located on both sides, enclose the pit chamber (lumen) with outgoing edge bulges, emanating from the secondary cell walls. In the middle of this cavity, the margo and torus are located (Figure 1). The edge bulges leave the pit chamber partially open through a pore so that media can flow freely through the membrane when open. The margo consists of elastic microfibrils and allows the plate-like torus to shift under pressure and close the pores. This way, unusual pressure differences inhibit unintentionally large amounts of medium from flowing freely [12,13,14,15,16,17,18].

## 2. Materials and Methods

### 2.1. Design Space

Following the top-down biomimetic design process [19], the requirements of the electro-automotive cooling system are applied to the biological search space, and the best suited model is determined. Then, the biological model is separated into its functional elements and reproduced in technical designs. These are evaluated for functionality as well as the ease and cost effectiveness of production. A converged design then undergoes consecutive development by modeling and verification approaches.

### 2.2. Finite Element Modeling

The first selection of the technical pit designs is then evaluated and subsequently optimized using finite element methods (NX Simcenter Nastran/CFD, SOL601 and 106) to look for (i) linear and predictable closing movement and (ii) the adjustable pressure (adjustable by dimensions other than diameter of the whole valve (thickness, spiral length, and radius)) necessary to close and/or open the valve. All simulations are performed with Simcenter Nastran using the solution types SOL601 and 106 for a material thickness of 0.5 mm to 1 mm and various polymers at pressures between 0 Pa and 70 Pa. These conditions equal a wind speed of approximately 10 m/s or 35 km/h, i.e., when the switching behavior is most desired (Figure 2).

### 2.3. System Verification

The final pit configuration is integrated into a cooling subsystem fitted to an existing design of a series production car model and subsequently prototyped. Rigid elements of the system as well as testing elements are produced with Prusament PLA filament on a Prusa MK3S+. Flexible elements like membranes are made on the same printer but with thermoplastic polyurethane Z-Semiflex and an added drying procedure.

Employing varying combinations of air ducting allows for fine-tuning the most optimal dimensions for the model application. Air pressure is generated with an array of six fans (3214 JN, ebm-papst, Mulfingen, Germany). Air speed is measured with a AF 210 hot wire anemometer (Digitron, Ferentino, Italy).

## 3. Results

### 3.1. From the Biological Model to a Technical Concept

Feasible technical design can be very far from the biological model since materials and production methods decidedly differ (Figure 3). In the attempt to maintain strong causality between design choices and resulting behavior, the redundancies of the biological model are removed while maintaining their functional principles:

(i) The pit is reduced to a single pit, which is embedded in a circular frame (Figure 4a). The frames of all pits are directly integrated into an overall structure and form a comprehensive area [20]. (ii) The large number of thin microfibrils connecting the central torus plate to the frame are replaced by a few large arms (Figure 4b). (iii) The closing torus plate held by the arms in the middle of the spot changes in shape from a flat ellipsoid to a perfect circular design. (iv) The flow space of the dot—the lumen—keeps its ellipsoid shape. (v) In contrast to the torus, the lumen is hollow to initially create a flow space. The lumen has a hole opposite to the torus. (vi) This pore is as large as possible to maximize air flow, so that the torus is just about able to close it completely.

The mechanical requirements of mobile and static structures differ considerably and hence a multi-part solution was chosen: The flexible matrix mimicking the pit membrane is located in the functional structure (Figure 4b; green). Together with the connecting structure, the membrane forms a two-dimensional mat. Antagonistic to this is the rigid support structure forming the entire lumen recess and pores. The structure is further divided into individual modules to increase the potential design parameter space. The hexagonal shape allows not only for an efficiently packed, two-dimensional arrangement but—by applying tiny shape adjustments at the edges—for three-dimensional curvatures similar to fullerenes (Figure 5). The size of the elements is adjusted to the dimensions of a typical wheel arch cooling element providing easy geometric matching for the selected use case.

### 3.2. Mechanical Behavior

The application of the pit valve requires predictable and linear movement, and to configure the closing pressure and (lack of) leakage.

The structural design was focused on generating a controlled, linear, and uniform movement of the torus plate along the z-axis towards the opening. This was achieved best by a two-arm design leading to far more uniform behavior compared to the attempted one-arm design, albeit with a visible rotational component around the z-axis. As a result, an even closure of the opening was observed (Figure 6).

The behavior of the torus plate changed considerably with the variation in its thickness (Figure 7). While the closing of the thick membrane design happened uniformly, the thin membrane design closed in an s-shaped fashion. In the simulation, a displacement of 8 mm was needed to achieve full closure and with it, air flow blockage.

In the chosen application case, a pressure of 70 Pa was derived as the desired closing pressure as this would equal roughly 30 km/h, a critical transition where aerodynamics become increasingly important in fuel efficiency. At a given displacement, the closing threshold can be achieved at different pressures by adjusting the membrane thickness (Figure 8).

A stress analysis was performed on the structure. Under the extreme displacement of the torus plate of 20 mm (instead of the 8 mm design displacement), weaknesses of the design and breakpoints became visible. In the simulation (Figure 9), the stress was rather uniformly distributed, with moderate peak stress at the end points of the spiral arms, both at the base and the torus plate insertion.

The sealing properties of the valve were influenced both on the encasing and the torus plate. The variants simulated exhibited a 45° angle at the contact point between the encasing and torus plate. All simulation pressures were in the working range of the application case and hence no deformation of the torus plate was observable. (Figure 10). 

### 3.3. Material Selection

Upon deflection, the membrane experiences a multi-axial stress state consisting of bending and torsion. The deformations of these are added via superposition. In simplified terms, the spiral arms are assumed to be a clamped linear bending beam whose length can be determined by the function of the Archimedean spiral used. The torsion depends on the distance to the center of the plug, which can also be determined mathematically. The stresses occurring in the spiral arms of the membrane are very low due to their small thickness. The deformation is therefore in an approximately linear elastic range (Figure 11).

### 3.4. Verification

The first verification tests showed incomplete closure of the valve caused by surface roughness and stiffness of the membrane (Figure 12a). Therefore, firstly, the counter-structure received a tiny circular thickening (Figure 12b,c) such that the membrane came to rest and a complete closure was achieved. Secondly, the membrane was fitted with a tiny circular thickening (Figure 12b), which aided in the reliable axial positioning and hence the closure of the valve. Both changes were verified in the setup. The airstream was now fully blocked (Appendix A). In terms of manufacturing, a combination of (b) and (c) appear most feasible, as the flexible—and here completely flat—structure can simply be punched from roll material. Additionally, the lower mass aids in valve responsiveness.

The air duct for the chimney effect is mounted above the pit matrix (i.e., a two-dimensional array of pits) directly on the wheel arch radiator, at the highest point directly above the cooling elements. The chimney is attached sideways due to the confined space. The pit elements are used in valve closing at increased external pressures resulting from either active fan cooling or driving-induced winds. The chimney allows for hot air to leave the cooling system passively and silently without interfering with the other cooling systems (Figure 13).

## 4. Discussion

Here a biomimetic valve system derived from the bordered pit found in higher plants is presented. This system is highly tunable by design, cost-effective to produce, noise-free, and it functions autonomously. It allows for adaptive air flow management in electric vehicles, bypassing the back-pressure-based systems at low driving speeds, and hence facilitating cooling in the absence of active fanning.

The basic design as derived from the biological model was modified to fit production methods and materials, e.g., the modification of the margo, from microfibrils to the spiraling arms as moving parts, while the torus plate and pores remained close to the biological model. The one-part model developed into a multi-part design space where the individual elements were easy to produce.

This initial technical design leaves space for future design choices of the individual components. In all possible variants, the basic function is the same: a valve that can open and close, passively driven by flow forces or pressure differences, respectively. The design and geometry of the arms are most influential for applications in which the technical pit can be used.

### 4.1. Parameterization

The modular design allows for high adaptability to all kinds of application cases. In consequence, we developed a design formula to aid in dimensioning only through the adjustment of geometric parameters (Figure 14). Additional influence was exerted by the Young’s modulus of the material used in the specific application case. Next to thermoplastic polyurethane, metals and ceramics are also possible [22,23].

The geometric and material choices strongly influenced the behavior of the structure and hence allowed for focused tuning. In short, the relationships were as follows: (i) A longer arm became softer, (ii) a thinner arm was more influenced by gravity, and (iii) a thicker arm moved more linearly (the behavior of the torus plate changed considerably with variation in its thickness (Figure 7). While the closing of the thick membrane designs happened uniformly, the thin membrane design closed in an s-shaped fashion. In the simulation, a displacement of 8 mm was needed to achieve full closure and with it, air flow blockage. (iv) The stiffness of the system increased linearly with the number of arms, (v) there was a cubic influence of membrane thickness, and finally, (vi) the Young’s modulus of the material directly influenced the system performance. The precise calculations for dimensioning were highly specific to the setting and material used, and hence could only be made using a detailed finite element analysis or similar methods. These empirically developed relationships allow for quick estimation in the designing process.

### 4.2. Chimney Effect

The biomimetic valve allows for free air flow around the vehicle´s cooling system, bypassing the back pressure system common in modern automotive aerodynamics. Since warm air has a lower density than cold air, hot air rises in a connected system. If this upswing takes place in a vertical, enclosed space, with openings only at the top and bottom and a heat source at the lower opening, then the upward mass flow draws new air from below to the upper opening. This naturally occurring convection is called, among other names, the chimney effect [24]. This effect has already been used in traditional Middle Eastern architecture, but is also considered in contemporary architecture and technical applications. Likewise, this type of natural convection is used in the constructed habitations of animals. Applied to a modern vehicle, a chimney combined with the biomimetic valve presented here would allow for the generation of fan-free, and hence cost-, energy-, and noise-free cooling air flow (Figure 15). This is highly beneficial in electric vehicles, where cooling is needed at standstill (when charging) and at low speeds (when most momentum is generated)—most contrary to cooling demands of classic carbureted vehicles. The physical requirements for the sufficient generation of air motion calls for a chimney of 1 m or longer; hence, it would need to be integrated into the design of the most adjacent roof pillar.

### 4.3. Features and Industrial Application

The pit matrix shows interesting features supporting its industrial application:

**Fewer parts.** Depending on the fabrication method, the valve can be produced from one to two parts by using materials of different flexibility. The simple geometry of the bending object allows for the usage of standard production methods. Due to the clear design, the assembly process is less demanding [26].

**Predictable movement and stress.** Its few components make the movement of the flexible parts highly predictable. Lubrication is unnecessary, as minimal friction occurs only between moving parts in the closed state. Likewise, the prediction of wear due to the deformation of a flexible membrane is quite easy, with the appropriate software [27].

**Parameterizable.** As mentioned above, the modular design and the operating principle allow the pit valve to be easily tuned for many different pressure ranges [28].

**Weight.** The few components, the flat design, and the material used make the pit valves particularly light. The pit matrix weighs only 100 g in the version presented here made of solid polypropylene. Further savings can be achieved by adapting the internal structure.

**Price.** The production of the matrix is cheap compared to other valves [29], On one hand due to the few parts and the resulting simple fabrication, and on the other hand due to the low weight and the cheaper material. The pit matrix can be used for low pressures and is able to be designed for any pressure by adaptation. The parameterizable properties include the geometry of the membrane, the material thickness, and the material strength. The valve itself is therefore also suitable, in principle, for higher pressures.

Consequently, a range of different uses for the pit valve is possible:

**Temperature-dependent valve control.** The general functionality of the pit valve can also be applied in other ways. If the modulus of elasticity is tailored to the temperature such that the torus’s dead weight cannot be sustained at a critical threshold, then a valve responsive to temperature could be developed. Depending on its orientation, this valve would open or close at a critical temperature. Such a design is especially advantageous for cooling systems where continuously open valves prove disadvantageous or undesirable during colder temperatures.

**Pit on both sides.** When placing closing pores on both sides of the torus, the pit valve can be tuned towards pressure ranges, i.e., not only does it close if the pressure is too high, but the dead weight of the torus location also allows for keeping the valve closed if the pressure is too low.

**Area of application based on biological model.** In trees, the pit serves to regulate the pressure of liquid media and control for foreign objects in the flow [30]. The current proposed design of the pit valve can do just that. Thus, the valve may be installed in a stream to allow for flow at low flow velocities. However, should the pressure in the stream increase unintentionally, the valve would close and no longer open until the pressure drops again. It would also stop unwanted debris from flowing through the vent.

**Use for extreme pressure ranges.** Depending on the choice of material and geometry, it is possible to use the presented valve in a wide range of pressures from a few Pa to several MPa, and probably beyond. Especially in the aerospace industry, it is of great importance that components are as light as possible. Consequently, it would be a conceivable application to design a high-pressure valve made of titanium, which possesses space-saving geometry and is lightweight. A specific application could be a safety valve in the event of sudden pressure loss due to damage to the outer surface of aircraft or space stations. Especially with regard to the increasingly filling orbit, the probability of possible damage to spacefaring objects increases. Comparable titanium bending modules are already in use at NASA [31,32].

## 5. Conclusions

The result of this work is a flat, passive valve, whose flow control function is carried out by elastic bending. Due to the bending, there are advantages over other valves. The valve, in combination with the chimney, generates flow through natural convection, helping to provide passive cooling within an existing cooling system. The generated air flow is clearly below active solutions, but is still perceptible, especially in cases where the chimney can be extended. As a result, the new cooling system can be used as a complementary technology in combination with existing cooling systems, keeping system complexity low. Especially in cases of low speeds, where it is not possible to cool via the airstream, the pit valve chimney can make the activation of the fan obsolete. Especially during battery charging, the cooling capacity of natural convection through a chimney could be sufficient. This allows the pit valve chimney to fulfill its intended purpose of noise reduction and power saving. With the rise of renewable energy, the need for stationary batteries is also increasing, and so is the need for cooling.

## 6. Patents

The results are protected under a German patent registration DE102022101791A1 [25].

## Figures and Tables

**Figure 1 biomimetics-09-00207-f001:**
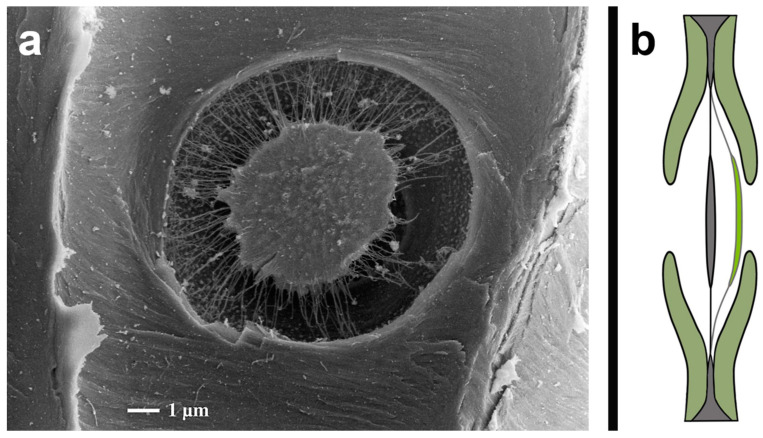
(**a**) Microscopic image of a bordered pit (with kind permission of Prof. Dr. Gerhard Wanner). (**b**) Drawing of an opened (grey) and closed (green) bordered pit (drawing by Anastasiya Mironava based on [15]).

**Figure 2 biomimetics-09-00207-f002:**
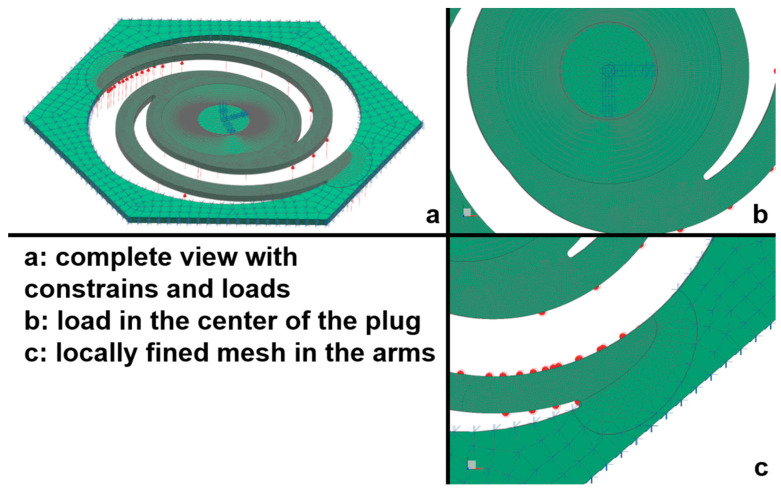
FE mesh on biomimetic bordered pit valve. Meshing of arms and arm-adjacent base is highly refined to handle expected complex deformations in those areas.

**Figure 3 biomimetics-09-00207-f003:**
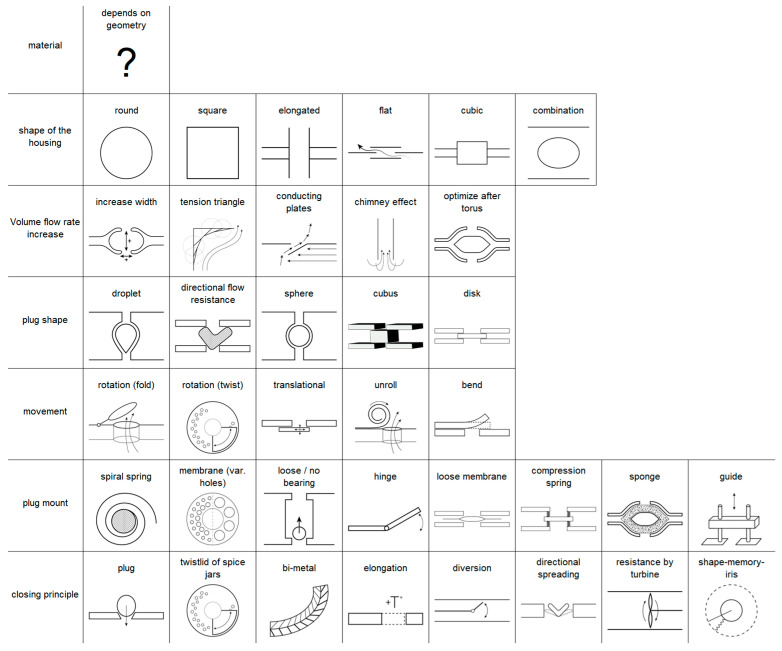
Technical design space. Based on the biological model, a functional technical design needs to also address requirements like desired function parameters, material, manufacturability, and cost (drawings by Anastasiya Mironava).

**Figure 4 biomimetics-09-00207-f004:**
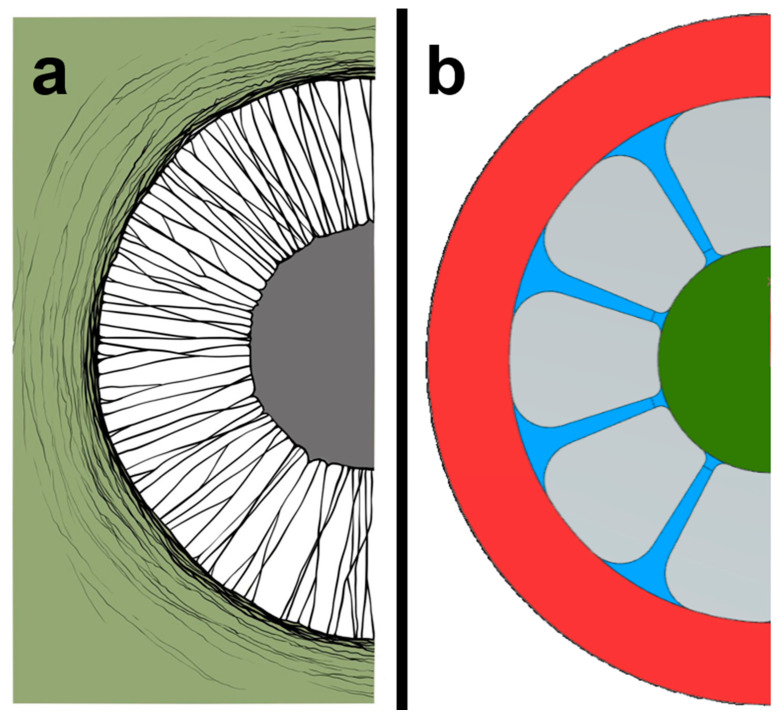
(**a**) Drawing of the biological model (drawing by Anastasiya Mironava based on [15]). (**b**) Technical abstraction of the bordered pit. Highlighted in red: frame. Red: base. Blue: microfibrils. Green: torus plate.

**Figure 5 biomimetics-09-00207-f005:**
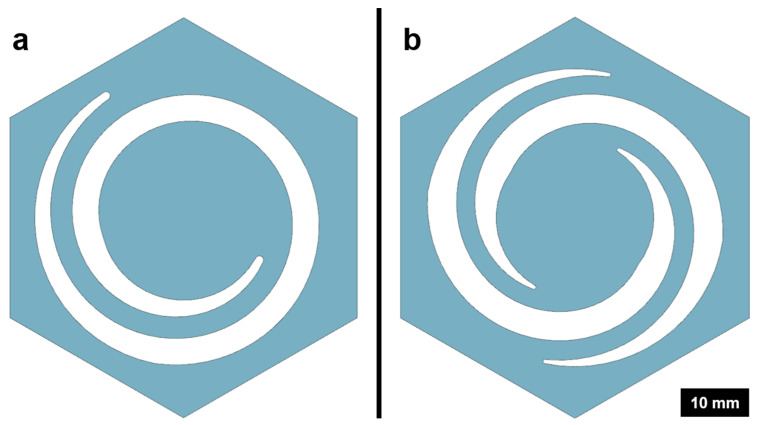
Design variants influence closing behavior. Membrane with one arm twists while closing (**a**). Optimized membrane with two arms (**b**) achieves symmetrical closing and hence higher process repeatability.

**Figure 6 biomimetics-09-00207-f006:**
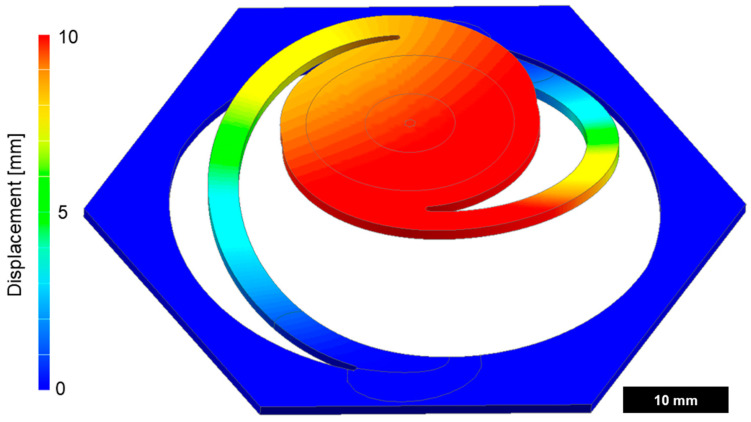
Mechanical simulation of torus displacement. The plate has a thickness of 1 mm, a diameter of 22 mm, and a Young’s modulus of 35 MPa. Pressure is applied from below. The displacement is accompanied by a counterclockwise rotation of the torus.

**Figure 7 biomimetics-09-00207-f007:**
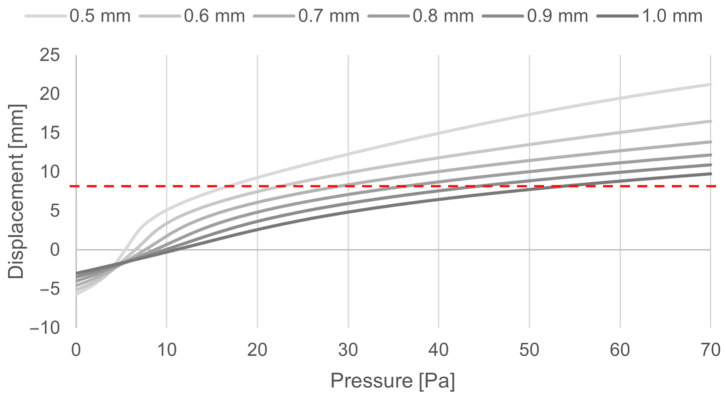
Displacement of the torus as a function of pressure and thickness. Closure is complete at 8 mm in the current design (red dashed line) while the total height of the valve is 10 mm.

**Figure 8 biomimetics-09-00207-f008:**
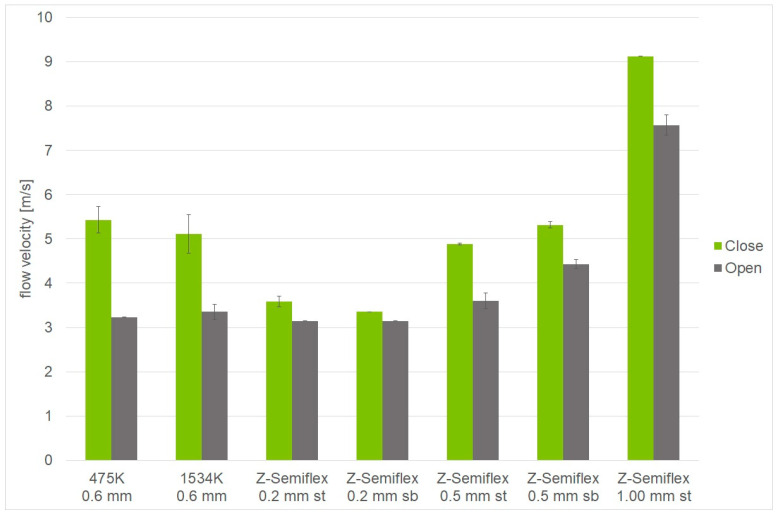
Opening and closing behavior depending on material used. Membranes tested were made from different materials (475K, 1534K, Z-Semiflex), in different thicknesses (0.2, 0.5, 1.0 mm), and mounted in different orientations, e.g., with the air-flow-exposed bottom being a smooth surface (sb) or the opposite, top side being smooth (st). In all cases, the encasing was 20 mm-thick and an exhaust chimney was situated on top of the setup.

**Figure 9 biomimetics-09-00207-f009:**
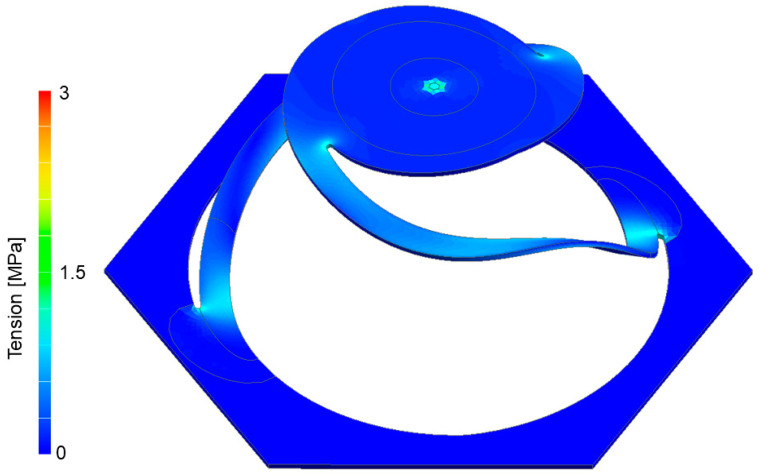
Plate displacement along a linear line. Despite the extreme displacement of 20 mm stress within the model, it remains fairly low and uniformly distributed along the structure. Peaks only occur at the end points of the spiral arms (stress point at the center of the torus plate results from force constraints during simulation).

**Figure 10 biomimetics-09-00207-f010:**
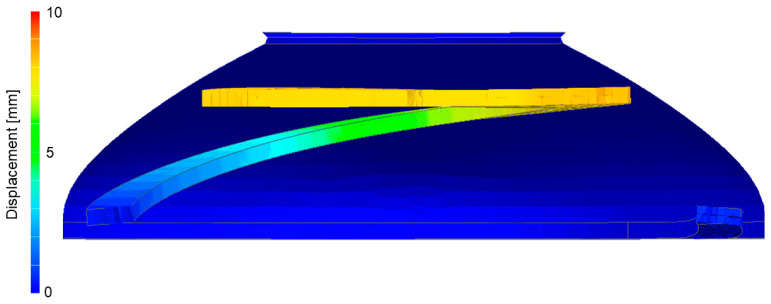
Simulation of plate displacement inside a rigid encasing. At 8 mm displacement, contact between both elements was achieved. Top 2 mm is dead space to compensate for tolerances.

**Figure 11 biomimetics-09-00207-f011:**
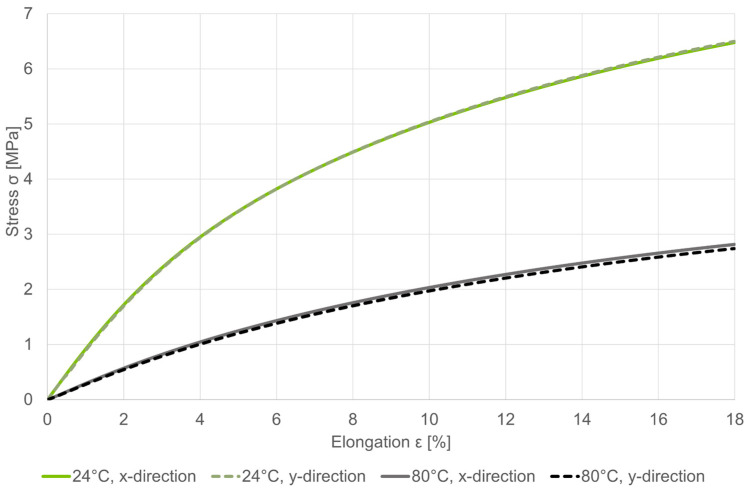
Stress–strain behavior of Z-Semiflex prototype material testing following DIN ISO 527-2. [21] Test specimen (type 1B) printed in massive material. Loading occurred at 100 mm/min; n = 6 experiments per direction.

**Figure 12 biomimetics-09-00207-f012:**
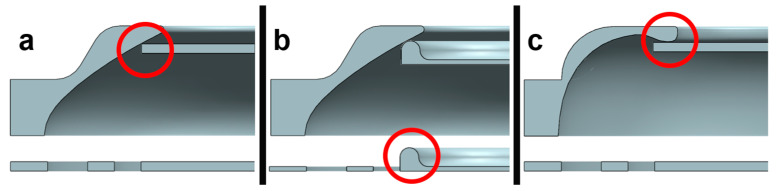
Controlling air tightness in closed configuration. While the original geometry (**a**) closes with a considerable leakage of air flow, the improved geometries (**b**,**c**) largely inhibit air flow. Variant (**c**) is costly to produce and hence (**b**) marks the final design.

**Figure 13 biomimetics-09-00207-f013:**
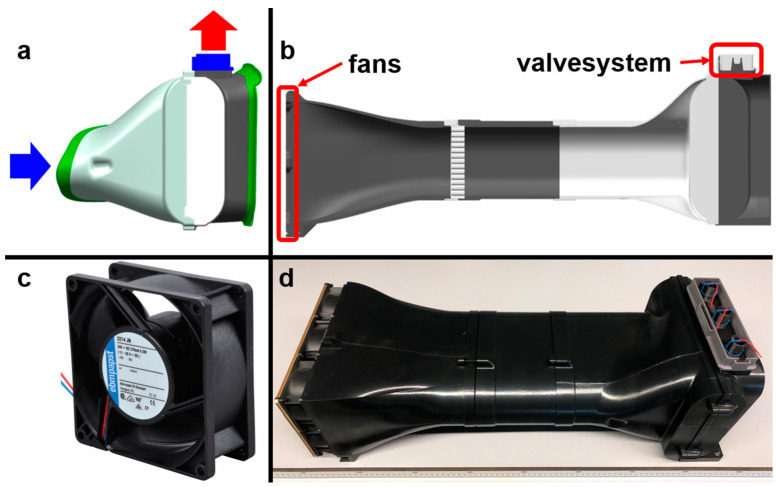
Setup for functional verification. For reliable verification, the use case is further narrowed down to a specific mid-range car model focusing on the forward right side. The front of the car is situated on the left side, and the front wheel on the right side of (**a**). In standstill, the valves are open and warm air exits the cooler at the top. (**b**) The existing design is extended with an air flow generator and—for manufacturing reasons—further modularized. (**c**) Air motion is generated by an array of 12 variable fans. The valve array is placed above the right-hand wheel arch cooler (**c**,**d**) and permits or prevents upwards air flow. Air flow is visualized by red wool threads in the final setup ((**d**), top right corner).

**Figure 14 biomimetics-09-00207-f014:**
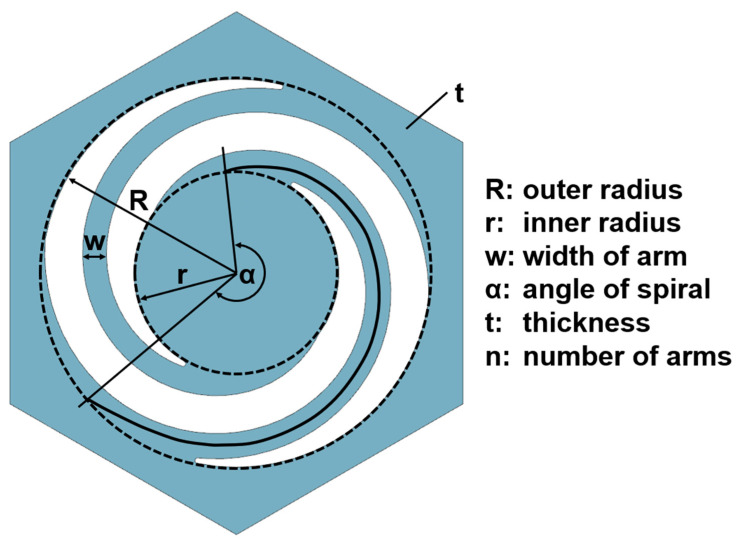
Geometric design parameters used to adjust the behavior of the torus.

**Figure 15 biomimetics-09-00207-f015:**
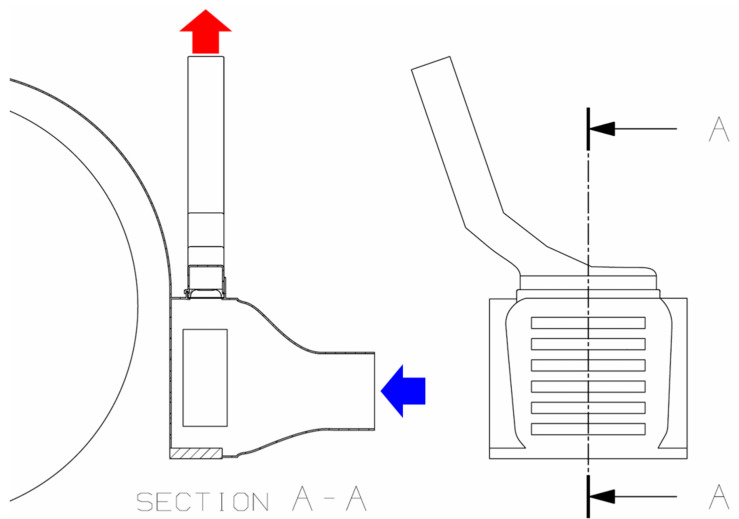
Abstract design of chimney effect adjusted to the wheel arch cooler setup. Air flow with open valves is indicated by red and blue arrows. The chimney shape is a result of geometric constraints given in the application case [25], [adapted].

## Data Availability

The datasets generated and analyzed during the research are available from the corresponding author upon reasonable request.

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
