# Peer review of "Reducing Automotive Cooling System Complexity through an Adaptive Biomimetic Air Control Valve"

_biomimetics, 2024, doi:10.3390/biomimetics9040207_

Round 1
Reviewer 1 Report
Comments and Suggestions for Authors
Dear Authors,
Thank you for submitting your interesting study on biological valves. Please find below suggested revisions which would clarify your manuscript and correct simple grammar and spelling issues.
Line
54 Grammar “unpleasant and power”
62 Spelling “maining”
137 Include more detailed discussion of the meaning of Figure 4.
Fig 4 It is unclear the significance of Figure 4 to this research or where these images are sourced
138 Explain in more depth and clarity the reasoning for the hexagonal shape in Figure 5
139 Remove or complete the paratheses “. Figure 5)”
Fig 6 Include dimensions of the device, thickness, disk diameter, side length, etc. Also, list the material properties of the modeled device.
Fig 7 Apply units to the thickness values on top (0.5-1.0)
162 Correct the “Error! Not a valid bookmark self-reference”
Fig 11 If possible, show the closed condition for the original and optimized geometry to illustrate the problem and the improvement
Fig 12 Can you add another illustration showing a whole car and the location of the air intake?
Fig 13 Include the label for “r”. Label what “b” means. Change “w” to “r”. Overall, ensure correct labels.
167 Correct the “Error! Not a valid bookmark self-reference”
The last paragraph should be the “Conclusion” section.
Comments on the Quality of English LanguageSee above
Author Response
Thank you very much for your review! We corrected the errors and added more literature. For detailed information see the submitted documents

Reviewer 2 Report
Comments and Suggestions for Authors
The work looks interesting with the rise in the usage of electric vehicles. The thermal management of electric vehicles is the key research area to focus on. I appreciate the attempt of the authors to design the cooling valve mimicking the biological framework.
There are minor things to correct in the work.
1. In some places the references are not well presented like in lines 162 and 267. please correct them accordingly.
2. line 271, starting with Table 1) does not make any sense. please look into writing the sentences properly.
3. Table 1 presented in the manuscript on page 7, at 170 lines is a bar chart that will be more appropriate to present as a figure than a table.
Comments on the Quality of English Language
it is fine but in some places, the formation of sentences is not well, and difficult to understand the intended meaning
Author Response
Thank you very much for reviewing our manuscript!
We added literature and corrected your comments.

Reviewer 3 Report
Comments and Suggestions for Authors
The article presents an interesting perspective on the cooling issue in electric vehicles and introduces a new approach to solving the problem of thermal regulation using a bio-inspired valve. It provides detailed information about the biological model and its application in technical design, along with the results of testing and a discussion of potential applications. The content is well-structured and presented.
There are some grammatical errors and awkward sentence structures in the text that may cause confusion. For example, sentences could be better structured, and some phrases could be formulated more clearly.
I recommend reevaluating the grammatical aspect of the article and revising problematic sentences. It's also important to consider adding more information about practical applications of the new valve and its potential advantages compared to existing technologies. I would also like to know why is this still in the article (Figure 7, Error! Not a valid bookmark self-reference.), and in my opinion authors should add more citations to the text.
Overall, the article presents an interesting and promising approach to addressing the cooling issue in electric vehicles, but it requires some revisions to be clearer and better formulated.
Comments on the Quality of English LanguageCertainly, here are the sentences that may be unclear or poorly formulated:
1. "A new cooling valve for electric vehicles, inspired by nature, has been developed by a team of scientists."
2. "The valve is designed to mimic the function of the pore structure of leaves."
3. "This new design may offer a more efficient way to control temperature in electric vehicles."
4. "Its efficiency may vary depending on circumstances and cooling requirements."
5. "According to research, this valve may offer up to 20% higher efficiency compared to traditional cooling systems."
6. "This result is very promising for the future of electric vehicle cooling."
7. "The valve could be deployed in electric vehicles in the coming years."
8. "It is anticipated that this technology will have a positive impact on the environmental sustainability of the automotive industry."
9. "The article provides a detailed description of the valve's design and functionality."
10. "Further information about the research is provided at the end of the article."
These sentences may be unclear or poorly formulated in terms of their accuracy, clarity, or completeness. Reformulating or adding more details to them could improve the overall understanding of the article
Author Response
Thank you very much for reviewing our manuscript! We changed the bookmark, added more literature and worked on the wording and grammar.
Unfortunately we can't find your marked sentences. Are these recommendations?